# Tackling the Behavior of Cancer Cells: Molecular Bases for Repurposing Antipsychotic Drugs in the Treatment of Glioblastoma

**DOI:** 10.3390/cells11020263

**Published:** 2022-01-13

**Authors:** Michele Persico, Claudia Abbruzzese, Silvia Matteoni, Paola Matarrese, Anna Maria Campana, Veronica Villani, Andrea Pace, Marco G. Paggi

**Affiliations:** 1Cellular Networks and Molecular Therapeutic Targets, Proteomics Unit, IRCCS-Regina Elena National Cancer Institute, 00144 Rome, Italy; mpersic1@bidmc.harvard.edu (M.P.); claudia.abbruzzese@ifo.gov.it (C.A.); silvia.matteoni@ifo.gov.it (S.M.); 2Center for Gender Specific Medicine, Istituto Superiore di Sanità, 00162 Rome, Italy; paola.matarrese@iss.it; 3Department of Environmental Health Sciences, Mailman School of Public Health, Columbia University, New York, NY 10032, USA; amc2465@cumc.columbia.edu; 4Neuro-Oncology, IRCCS-Regina Elena National Cancer Institute, 00144 Rome, Italy; veronica.villani@ifo.gov.it (V.V.); andrea.pace@ifo.gov.it (A.P.)

**Keywords:** glioblastoma, drug repurposing, drug repositioning, antipsychotic drugs, review

## Abstract

Glioblastoma (GBM) is associated with a very dismal prognosis, and current therapeutic options still retain an overall unsatisfactorily efficacy in clinical practice. Therefore, novel therapeutic approaches and effective medications are highly needed. Since the development of new drugs is an extremely long, complex and expensive process, researchers and clinicians are increasingly considering drug repositioning/repurposing as a valid alternative to the standard research process. Drug repurposing is also under active investigation in GBM therapy, since a wide range of noncancer and cancer therapeutics have been proposed or investigated in clinical trials. Among these, a remarkable role is played by the antipsychotic drugs, thanks to some still partially unexplored, interesting features of these agents. Indeed, antipsychotic drugs have been described to interfere at variable incisiveness with most hallmarks of cancer. In this review, we analyze the effects of antipsychotics in oncology and how these drugs can interfere with the hallmarks of cancer in GBM. Overall, according to available evidence, mostly at the preclinical level, it is possible to speculate that repurposing of antipsychotics in GBM therapy might contribute to providing potentially effective and inexpensive therapies for patients with this disease.

## 1. Introduction

### 1.1. State of the Art in Glioblastoma Therapy

Glioblastoma (GBM, grade IV astrocytoma) is the most common primary malignant brain tumor in adults [1], with a median age of 65 at diagnosis [2]. GBM is well-known for its poor prognosis, with a median overall survival of 14.6 months, even using the conventional, state-of-art first-line treatment, i.e., surgery followed by radiotherapy (RT) plus temozolomide (TMZ) (also known as the “Stupp regimen”) [3]. Currently, several drugs are utilized for recurrent GBM after first-line treatment, e.g., bevacizumab, regorafenib and nitrosureas, but no standard treatments have been clearly identified for progressive disease [4,5].

Moreover, the clinical benefit of Stupp protocol is correlated with the methylation status of the *MGMT* gene, a strong predictive biomarker of the response to temozolomide chemotherapy; indeed, only GBM patients with high methylation levels of *MGMT* have a survival advantage from this treatment [6]. Thus, there is a great need for new research exploring the activity of new drugs, particularly in GBM patients carrying the hypo- or unmethylated *MGMT* gene.

Overall, such a poor patient prognosis is the consequence of some GBM features: (a) its highly invasive nature [7], which is responsible for the persistence of tumor residues after surgery, also undetectable on MRI [8]; (b) the presence, within the neoplastic lesion, of different cellular subsets deriving from a complex hierarchy of progenitors and glioma stem cells (GSCs) [9,10,11]; (c) the ability to develop functional multicellular network structures able to invade the surrounding parenchyma and repair interconnected damaged cancer cells [12].

When compared with the previous gold standard in GBM therapy, the Stupp regimen allows an increase of 2.5 months in overall survival [3], which, albeit remarkable, remains unsatisfactory. Therefore, novel therapeutic approaches and effective medications are highly needed and incessantly sought.

### 1.2. Drug Repurposing in GBM

Since the development of new drugs is an extremely long, complex and expensive process [13,14], scientists and clinicians are strongly motivated to consider drug repositioning/repurposing as a valid alternative [13,15], especially now that novel bioinformatics and multi-omics platforms can contribute in assessing the potentialities of old and well-known drugs, as well as those used in other diseases. Indeed, drug repurposing/repositioning, when supported by solid mechanistic rationale and straightforward preclinical data, can overcome many issues related to preclinical and clinical testing [16,17]. Furthermore, repurposed drugs may provide other benefits, including (a) safety, because data on doses and toxicity are already available [18,19]; (b) cost, because they are less expensive than newer drugs [14,19]; (c) ability to reach the bedside faster [14].

Drug repurposing is also under active investigation in GBM therapy, since a wide range of noncancer and cancer therapeutics have been proposed or investigated in clinical trials [18,20,21]. Among these, a remarkable role is played by the antipsychotic drugs (also known as neuroleptics or major tranquilizers), an issue surely connected not only with their known mechanism of action (MoA) but also with other partially unexplored interesting features [22,23,24].

### 1.3. Repurposing Antipsychotics in GBM

A key role of antipsychotics in cancer therapy is also supported by studies that demonstrate a reduced cancer incidence among patients affected by schizophrenia taking neuroleptic medications [25,26,27,28,29,30,31,32] and which, in addition, report sporadic data showing a better course of GBM in psychiatric subjects taking neuroleptic drugs [25,33].

Indeed, antipsychotics are a class of psychotropic drugs used for the treatment of bipolar disorder, psychosis, delirium, Huntington disease and Tourette syndrome. These drugs are not devoid of well-characterized side effects, such as sedation and extrapyramidal syndrome, most of them dose-dependent and reversible upon drug withdrawal. Based on the risk of developing extrapyramidal symptoms (EPS) and tardive dyskinesia (TD), antipsychotics are divided into typical or first-generation antipsychotics (FGA), and atypical antipsychotics, which include the second-generation ones (SGA) (Table 1).

Antipsychotics, according to their known MoA, can bind with different affinity and modify the activity of several receptors of neurological interest, as dopamine receptors (DR), muscarinic M1 receptors, α-adrenergic receptors, serotonin (5-HT) receptors and histamine H1 receptors [34]. The reduced risk of EPS and TD of the SGAs is attributed to the stronger 5-HT_2A_ receptor antagonism, compared to DRD2 antagonism, and faster dissociation from DRD2 [35].

Conventionally used typical antipsychotics include phenothiazine derivatives (e.g., chlorpromazine) and butyrophenones (e.g., haloperidol). Since atypical antipsychotics such as clozapine (CLO), risperidone (RIS) and aripiprazole have a lower propensity to induce extrapyramidal symptoms usually observed using typical antipsychotics, they became more popular for treating schizophrenia patients. However, several clinical trials for patients with chronic schizophrenia have suggested a limited advantage of the newer agents in terms of efficacy [36,37,38].

First-generation antipsychotics are subdivided into high-, mid- and low-potency drugs, based on the doses needed in order to reach the desired clinical effect [39,40,41]. All antipsychotics are lipophilic substances with high permeability through the blood–brain barrier (BBB) [42], but there are still differences in their capability of reaching specific CNS regions [43]. Information about antipsychotics’ known MoAs, clinical indications and major side effects is summarized in Table 1.

## 2. Purpose of the Review

In 2000, Hanahan and Weinberg described six pivotal traits that drive malignant growth in virtually all cancer genotypes: sustaining proliferative signaling, evading growth suppressors, activating invasion and metastasis, enabling replicative immortality, inducing angiogenesis and resisting cell death [44]. After more than a decade, Hanahan and Weinberg updated their list by adding four further central driver alterations: avoiding immune destruction, tumor-promoting inflammation, genome instability and mutation, and deregulated cellular bioenergetics [45]. Antipsychotic drugs have been described to interfere at variable incisiveness with most of, if not all, these hallmarks of cancer, indicating how this pleiotropic class of medications can hinder cancer growth via several and possibly overlapping pharmacological MoAs. Herein, following the order and nomenclature reported in the 2011 Hanahan and Weinberg paper [45], we analyze the effects of antipsychotics in oncology and how these drugs can interfere with the hallmarks of cancer in GBM. To this end, we will examine key preclinical results describing the role that antipsychotics play in hindering the growth of GBM cells at the level of the 10 cancer hallmarks.

## 3. The Role of Antipsychotics in Hindering the Growth of GBM Cells at the Ten Cancer Hallmarks

### 3.1. Sustaining Proliferative Signaling

Cancer therapy is traditionally based upon delaying or impeding malignant cell growth, a phenomenon governed by a complex network of biomolecular events [46,47,48]. Antipsychotics interfere with several, often interconnected, signal transduction pathways significant for GBM growth and malignant features.

PI3K/AKT/mTOR pathway. This pathway plays a pivotal role in regulating cell survival, proliferation and differentiation [49], and is frequently upregulated in GBM [50]. CPZ and other cognate phenothiazines can inhibit in vitro the AKT/mTOR axis in malignant gliomas, thus hindering their pivotal biological mechanisms, from cell survival to mitosis [51,52]. Inhibition of the mTOR pathway, mainly at the level of the mTORC1 complex, is strongly interconnected with the activation of the autophagic process. How this last feature can influence specifically GBM survival will be described below.

RAS/MAPK pathway. The RAS/RAF/MEK/ERK signaling cascade is regulated upstream by growth factors and mitogens. It represents a pro-survival mechanism that controls gene expression and counteracts apoptosis [53,54]. This pathway is particularly active in glioma cells [55]. TFP and its derivative A4 [10-(3-(piperazin-1-yl)propyl)-2-(trifluoromethyl)-10H-phenothiazine] modify the expression of total and phosphorylated p38 and ERK in oral cancer cells, thus impairing their growth in vitro and in vivo [56]. Although EGFR inhibitors have failed Phase II clinical trials in GBM therapy [57,58], these molecules can be reconsidered in combination therapy with antipsychotics due to their effect on the MAPK pathway by the antagonism of EGFR and DR isoform D2, respectively [59]. Of note, considering the interplay between PI3K/AKT/mTOR and RAS/MAPK axes, a synchronous inhibition of both pathways appears essential for the efficacy of GBM treatment and avoiding the onset of drug resistance [53].

WNT/β-catenin pathway. This signaling pathway exerts pleiotropic functions in neurogenesis and neural stem-cell regulation [60], but also plays a pivotal role in GBM proliferation, motility and epithelial-to-mesenchymal transition (EMT) [61]. Indeed, thioridazine (THD), besides its influence on autophagy and apoptosis (see below), can attenuate the WNT/β-catenin signaling by inducing phosphorylation, and thus degradation, of the β-catenin molecule [23,62]. 

Neurotransmitters and neuromodulators. These molecules, such as γ-aminobutyric acid, serotonin, dopamine, glutamate and norepinephrine, play essential roles in brain physiology, from embryonic development to neuronal activity, by interacting with their receptors [63]. Recently, it was proven that neurons are connected to GBM cells via synapses that, when excited by the upstream neuron, enormously stimulate tumor growth, aggressiveness and migration in GBM cells [64,65], where glutamate, dopamine and 5-HT are the main mediators of these stimuli via their post-synaptic receptors [66,67,68].

Dopamine and 5-HT are fundamental for neurodevelopmental processes as well as neural stem cells and progenitor cell proliferation [63,69]. The ability of serotonin in influencing the MAPK and AKT signal transduction pathways [70], as well as the recently characterized involvement of neuromediators and their receptors in GBM genesis, growth and aggressiveness [64,65,66], provide a strong incentive for considering the use of antipsychotic drugs to counteract GBM growth to ameliorate patient prognosis. Phenothiazines elicit an antagonist effect on monoamine receptors, mainly DRD2, a feature that is considered the basis for their neuroleptic properties. However, this activity can be targeted to lower the dopamine- and serotonin-enhanced GBM metabolic rate, signaling and cellular plasticity [67].

Phenothiazines can also inhibit the α-amino-3-hydroxy-5-methyl-4-isoxazolepropionic acid receptor (AMPA receptor, AMPAR) [71], which was recently recognized as highly expressed in GBM and fundamental in driving its growth and progression [64,65]. Additionally, antidepressants and antipsychotics effectively inhibit the NMDA glutamate receptor [71], described as essential for nesting and proliferation of brain metastases from breast cancer [72].

Control of intracellular calcium amount and flux is essential for cell homeostasis, as metabolism and compartmentation of this ion are strictly regulated in normal and cancer cells [73,74]. Interference with calmodulin and other calcium-controlling processes is key to hindering GBM growth and malignancy [75]. Some neuroleptic drugs can bind to and inhibit the activity of calmodulin. Indeed, PMZ and THD are effective calmodulin antagonists [23], and FPZ irreversibly inhibits calmodulin activity in neuroblastoma and glioma cells, making these drugs promising candidates for GBM combined therapy [76]. TFP exerts a potent anticancer activity by binding and blocking calmodulin subtype 2, a factor highly expressed in GBM, which when inhibited generates cell toxicity by increasing Ca^2+^ intracellular concentration up to toxic levels [77]. Confirming the same trend, CLO inhibits the PI3K/Akt/GSK-3β axis via its inhibitory effect on calmodulin [78].

### 3.2. Evading Growth Suppressors

Cell cycle is governed by a continuous interplay between stimulating and suppressive stimuli, a setting where cancer cells can escape the effect of tumor suppressor genes and thus replicate relentlessly [79,80]. Qualitative and/or quantitative alterations in some cell-cycle checkpoints are also responsible for chemo- or radio-resistance in several cancers, including GBM [81]. Selected checkpoints are druggable, sometimes highly selectively, by specific molecules that can thus interfere with cell replication at different steps [50,82]. In this context, some antipsychotics, i.e., CPZ, THD, TFP and the phenothiazine derivative DS00329, may induce cell-cycle arrest in G1 [52,83,84,85,86]. On the other hand, other investigators report the ability of CPZ to block the cell cycle at the G2/M boundary [51,87]. Such discrepancies can be attributed to either a different MoA of each specific compound or a different cell histotype and culturing context.

### 3.3. Avoiding Immune Destruction

Immunotherapy has represented a breakthrough in the treatment of cancer. Results on clinical trials on immunotherapy in GBM have been quite disappointing since this tumor is immunologically “cold” and thus not responsive to anti-PD-1 or anti-PD-L1 monoclonal antibodies [88]. Nevertheless, a possible window is opening up through personalized vaccine therapies [89,90].

Glioma cells try to avoid the attack by the immune system by shutting down the antigen-presenting cells’ (APCs) activity of microglia [91], an effect mediated by IL-6 [92]. Antipsychotics, due to their ability in lowering inflammatory response, reduce the level of IL-6 [93], which could be effective in hampering immune evasion. In melanoma, another tumor from the neuroectodermal origin, the combined therapy employing PMZ plus specific siRNA-mediated PD-1 downregulation was found effective in vivo [94], while the combination of TFP with immune checkpoint blockade has been suggested to improve the antitumor efficiency [95]. In this last case, it should be outlined that monoclonal antibodiesdo not cross an intact BBB, so their effect should be restricted essentially to the extra-CNS immune compartment.

### 3.4. Enabling Replicative Immortality

In order to maintain an endless replicative capability, most cancer cells rely on telomerase reverse transcriptase (hTERT) activity to avoid excessive telomere shortening and the consequent block of the mitotic processes [96]. Telomeres are characterized by guanine-rich sequences that form the so-called G-quadruplex structures [97], and ligands that bind and stabilize G-quadruplexes interfere with hTERT activity and thus with cancer cell replication [98]. Incidentally, in GBM, the involvement of hTERT activity appears to affect mostly female patients [99]. Using a virtual screening procedure, prochloroperazine, promazine and CPZ have been identified as G-quadruplex stabilizers [100], but their effectiveness in vivo is still to be confirmed. In late-stage GBM evolution, hTERT promoter mutations generate an increase in telomerase activity [101], thus opening the possibility of using selected antipsychotics as hTERT inhibitors in combined cancer therapy.

### 3.5. Tumor-Promoting Inflammation

Inflammation is a critical disease modifier and has a well-documented role in tumor development [102]. CNS can undergo acute or chronic inflammatory conditions that can be considered potential triggers for gliomatous transformation [91]. A relationship has been postulated between the measles morbillivirus (MeV) infection and acute and chronic encephalitis that appears to promote glial transformation, often in the brain sites of infection and chronic inflammation [103].

In addition to onset, inflammation is also key in glioma evolution. Once glioma has developed, infiltrated microglia, responsible for the intratumor inflammatory state, release cytokines to promote cancer-cell growth [91]. Indeed, IL-1 and bFGF, released by microglia/TAMs, can induce tumorigenesis [104]. Although the glioma microenvironment is strongly immunosuppressive, M2-phenotype microglia and Treg, glioma cells still produce pro-inflammatory cytokines, such as IL-6 [105].

Antipsychotics are known as “fire extinguishers” in the context of a schizophrenic or psychotic brain [106] because they increase the levels of anti-inflammatory cytokines (IL-4 and IL-10) and suppress the levels of pro-inflammatory ones (IFN-γ) [107].

HAL and RIS, but not CPZ, can decrease the IL-6-dependent production of S100B, an insulin-related protein factor released in the context of neuroinflammation and upregulated in schizophrenic patients [108,109]. Hence, variations in the amount of S100B could reflect the effect of antipsychotics on neuroinflammation and, because this factor decreases in murine gliomatous models when treated with antipsychotics [110], it can be speculated that antipsychotics should possess anti-inflammatory properties in GBM patients by acting on S100B downregulation.

It is worth noting that promethazine, a phenothiazine derivative, is an old medication used since the 1950s that acts primarily as a strong antagonist of the histamine receptor H1 (antihistamine), as it is able to counteract the multiplexed functions of histamine and, among these, its role in inflammation [111]. Recently, anti-inflammatory therapy with histamine receptor H1 inhibitors was reported to enhance the effects of cancer immunotherapy by reverting macrophage immunosuppression [112].

### 3.6. Activating Invasion and Metastasis

Invasion and metastasis, key features of either local or distant cancer spreading, cannot be applied tout court to GBM clinical evolution. Indeed, GBM, due to its peculiar characteristics (e.g., BBB and brain as a “sanctuary” and/or necessity of specific neuromediators to sustain its growth), rarely metastasizes outside CNS. Nevertheless, its highly aggressive behavior and rich stem-like compartment are associated with a remarkable invasive capability. In addition, GBM cells activate EMT, a process devoted to increasing cell motility and plasticity [113] and can generate in vivo functional multicellular network structures via microtubules [12]. Such features set the ground, with almost no exceptions, for relapse and drug resistance [114], characteristics that strongly govern the dismal clinical course. There is a hierarchy of subpopulations in normal neural cells and GBM, which originates from the progenitor cells (apical pluripotent stem cells) and contains most of the cycling cells [11]. The GBM stem-cell status is strongly connected with the ability to replicate and the aptitude to migrate and invade the surrounding encephalic structures.

Some antipsychotics can interfere with the invasion and metastasis processes, possibly via the induction of a differentiation process [115]. Indeed, CPZ reduces cloning efficiency, neurosphere formation and downregulates the expression of stemness genes in neurospheres in GBM [87]. Fluspirilene suppresses proliferation and invasion in both GSCs (neurospheres) and GBM cells, acting via STAT3 inhibition [116]. STAT3 is a factor strongly connected with stemness, mesenchymal phenotype and resistance to RT and chemotherapy [117,118,119], which is also implied in cancer-driven immunosuppression [120]. On these bases, fluspirilene could be considered eligible for GBM treatment repurposing.

In constitutively active EGFR variant III (EGFRvIII) GBM, there is an increase in the activity of STAT3 and STAT5 [119]. STAT5 is involved in the increase of the expression of the TNFR family member fibroblast growth factor-inducible 14 (Fn14), a transmembrane protein that stimulates cancer-cell invasion and survival in GBM [121]. PMZ impedes GBM cells’ migration and survival by reducing the expression of Fn14 via hindering STAT5 activity [122]. The pattern of expression of the STAT signal transducers and transcription activators outlines that, while STAT3 is expressed in the core of the tumor, STAT5 is mainly found in the periphery, where it could play a major role in influencing local invasion [122]. Since local invasion is a noticeable obstacle to the clinical control of GBM, PMZ appears efficient in blocking GBM growth and could thus improve patient outcomes [23,122].

### 3.7. Inducing Angiogenesis

Elevated blood supply is essential for GBM growth, due to its noticeable energy supply and catabolite disposal needs. Clinically, the core of GBM displays necrotic areas due to the unreachability of this portion by sufficient blood flow by the pre-existing blood vessels; thus, neovascularization appears mandatory, even if often insufficient, for fully sustaining cancer growth.

The 5-HT receptor 7 (5-HT_7_) is highly expressed in malignant gliomas [23,123]. 5-HT stimulates this receptor and activates a number of oncogenic signals, among which is ERK1/2, leading to increased IL-6 production, with consequent STAT3 stimulation and VEGF synthesis. The latter is responsible for the increased angiogenesis in GBM, especially in the core of the tumor mass [123,124]. At present, three antipsychotic drugs, i.e., PMZ, paliperidone and RIS, have been shown to inhibit 5-HT_7_ effectively, thus downregulating ERK1/2, IL-6 and, ultimately, VEGF production and tumor vascularization [23,123]. Therefore, these three antipsychotic drugs may have a potential role in GBM therapy associated with the standard Stupp protocol.

### 3.8. Genome Instability and Mutation

Even if an elevated mutational burden can favor selecting cancer clones with increased malignant characteristics, cancer cells aim to reach a sort of genome stability to avoid an excess of lethal mutations.

Recent reports outline the ability of CPZ in inducing nuclear aberrations and ultimately mitotic catastrophe in GBM cells [83,87], while sparing normal neuro-epithelial cells [87,125]. Similarly, TFP decreases the expression of the homologous recombination (HR) proteins RAD51, BRCA1 and BRCA2 [126], thus leading to decreased DNA repair activity and, consequently, increased DNA damage. In this context, it should be outlined that the Stupp protocol for GBM therapy acts essentially by inducing genome damages via RT in combination with TMZ, an alkylating agent endowed with radiosensitizer properties, working thus in synergy with RT. Indeed, RT causes DNA double-strand breaks that must be repaired via non-homologous end-joining recombination (NHEJ) or homologous end-joining recombination (HEJ). RAD51, BRCA1 and BRCA2 play an essential role in HEJ [127,128], while TMZ, as an alkylating agent, also induces DNA damage synergistically. Here, the role of TFP in knocking down HEJ could be considered a possible explanation for its effect in enhancing the RT-induced toxicity [126]. Consequently, selected antipsychotics could be speculated to enhance RT sensitivity in patients with GBM, mainly in those repair-proficient that carry a hypo- or unmethylated MGMT gene promoter.

CPZ is an inhibitor of the mitotic kinesin KSP/Eg5, thus inhibiting tumor-cell proliferation through mitotic arrest and accumulation of monopolar spindles [83]. Moreover, in glioma cells, HAL causes mitotic arrest followed by inhibition of colonization in the scratch assay, proliferation and ultimately cell death [129]. A MALDI-TOF/TOF and 2D electrophoresis analysis showed changes in PRSS1, PCNT, PVALB, PRDX1, Rho GDI and GFAP protein expression after treating C6 rat glioma cells with RIS, HAL or CZP [130]. PCNT is an important scaffold for other centrosome proteins and plays a crucial role in mitotic progression. RIS and CPZ decrease the level of expression of PCNT in C6 glioma cells, whereas HAL does not [130].

### 3.9. Resisting Cell Death

Cancer cells acquire, via several mechanisms, an increased resistance towards death. One such tool is autophagy, in which older cellular structures are intensely recycled, mainly to induce a renewal of the cellular organelles and for energetic catabolic purposes. When in excess, autophagy, instead of representing a cytoprotective tool, turns out to become cytotoxic, bringing cells to death. Thus, autophagy-modulating compounds can generate imbalances in cancer cells, which are often characterized by high autophagic levels at baseline and, therefore, unable to increase the autophagy rate further.

CPZ triggers autophagy in the PTEN-null U-87 MG glioma cell line by inhibiting the AKT/mTOR axis, thus driving them toward a non-apoptotic cell death [51]. More recently, several other anchorage-dependent or -independent GBM cell lines (GSCs, neurospheres) were shown to undergo abortive autophagy when exposed to CPZ [87,125].

Autophagy might also be a major mechanism underlying the effects of THD on GBM neurospheres [22], since this drug is able to impair the fusion between autophagosomes and lysosomes, thus cooperating with TMZ in dysregulating the autophagic process [131]. In addition, THD enhances p62-mediated autophagy via the WNT/β-catenin pathway.

The protein p62 is versatile and is capable of regulating, in addition to autophagy, genetic stability, apoptosis and other forms of cell death in cancer cells. It acts as both a pro-oncogenic and a tumor suppressor protein by affecting proliferation, invasion, and response to chemotherapy and RT. Furthermore, p62 participates in activating or inactivating signaling pathways related to the tumor microenvironment and influencing EMT [132].

In addition, HAL reduces the level of Hsp70 in C6 rat glioma cells, also when pretreated with MK-801, a drug that increases the levels of Hsp70 [133]. Similar to HAL, RIS reduces the expression of Hsp70 [133], possibly leading to programmed cell death [134]. Downregulation of Hsp70 has been associated with cell death through a p53-independent mechanism, probably without the involvement of apoptosis, as suggested by the absence of DNA cleavage [135]. Therefore, HAL can possibly cause programmed cell death secondary to unfolded protein response (UPR), due to Hsp70 downregulation [134].

### 3.10. Deregulating Cellular Energetics

In the 1950s, Otto Warburg outlined the role of high glycolysis in cancer onset and progression [136,137,138,139]. Indeed, it has been postulated that cancers and malignant gliomas develop a “Warburg effect” to comply with the high energy metabolism activity for their anabolic needs [140,141,142]. Thus, approaches aiming to generate a bioenergetics imbalance by decreasing cancer cell ATP/ADP ratio are considered valuable to hit these cells, especially GBM [142]. However, the metabolic heterogeneity of this tumor and its overall plasticity make it difficult to pursue a univocal and decisive approach. In addition, metabolic imbalance affects GBM microenvironment and thus antitumor responses and immunotherapy outcomes [140,143].

Thioridazine is able to increase AMPKA and GSK3β activity in GBM cells [22,62], while CPZ and pimozide interfere with mitochondrial respiration in normal and cancer cells [23,144], thus subverting the metabolic equilibrium by decreasing glucose catabolism and increasing autophagy. Such an imbalance appears well tolerated in noncancer cells, while it creates a metabolic crisis in cancer cells, mainly GBM, ultimately driving them to death [145].

All the hallmarks of cancer described by Hanahan and Weinberg are recapitulated in Figure 1, where, alongside each hallmark, the antipsychotic drugs that can interfere with each individual trait are listed. This can potentially open up new possibilities for intervention in anticancer therapy. Interestingly, some drugs are able to interfere with tumor homeostasis through more than one MoA, which can be considered even more beneficial for their therapeutic efficacy.

## 4. Beyond the Hallmarks

Antipsychotics, besides the inhibitory effects described above on GBM and, in general, on cancer cells, display additional features worth describing in this context.

### 4.1. Neural Stem Cells Replication, Differentiation and Migration

Neural stem cells play a key role in CNS development and preservation, being the progenitors of neurons, oligodendrocytes and astrocytes. All these cells respond to monoamine neurotransmitters that are thus able to influence CNS homeostasis potently. Monoamines, mainly dopamine and serotonin, influence the proliferation, quiescence and differentiation status of neural cells [66]. All the drugs we describe in this article are functional in interfering with several monoamine/ligand interactions and thus able to interfere with neural stem cells and their homeostasis [146].

Preclinical and clinical data highlight how dopamine stimulates cellular proliferation within the brain [66] and, incidentally, outline how patients with Parkinson’s disease, usually dopamine-depleted, display lower incidence in brain tumors [28]. Consistently, experimental rodent models demonstrate that dopamine depletion decreases the proliferating capability of neural precursor cells in the subependymal and subgranular zone, and that their proliferation is restored via the administration of a DRD2 agonist. This strongly suggests that dopaminergic dysfunction could be accountable for the impaired generation of neural precursor cells in Parkinson’s disease patients [147].

In the developing and adult brain, the subventricular zone (SVZ) is a source of precursor cells for forming glia and neurons, and it plays an important role in GBM growth as well [148]. Via DRD3 signaling, SVZ cells undergo mitosis and centrifugal migration. Interestingly, GBM cells derive from these same progenitors and appear to take advantage of these monoamine-activated signaling pathways to grow and colonize the CNS successfully [149]. Incidentally, the mitogenic role of dopamine is considered responsible for the impairment of GBM cells to metastasize, due to the insufficient monoamine concentration available in the extra-CNS microenvironment [149].

Notably, atypical antipsychotics are more potent than typical ones at inducing neural differentiation [146]. It has been hypothesized that these drugs may affect chromatin configuration to recruit transcriptional activators or occlude repressors of neural differentiation machinery [150].

### 4.2. Neural Cells Survival Capabilities

CPZ shows a noticeable difference in toxicity between noncancer neuroepithelial cells and GBM cells. The former appears more resistant toward several effects elicited by this drug, e.g., nuclear fragmentation, mitotic catastrophe, ER stress and ROS production [87,125]. In the same direction, CPZ-induced autophagy, while playing a cytotoxic, abortive role in GBM cells, in neural cells appears survival-oriented, highlighting only in these noncancer cells the presence of key features able to turn the CPZ-induced autophagy into a resilient mechanism [125]. Similarly, three antipsychotic drugs, fluspirilene, TFP, and PMZ, are active in inducing autophagy and promoting long-lived protein degradation in neurons, thus proposing these compounds as useful tools in preventing cell death by neurodegenerative diseases characterized by the accumulation of misfolded proteins [151].

One key mechanism by which atypical antipsychotics confer neuroprotective effects is the modulation of oxidative stress. Several SGAs, including RIS, paliperidone, olanzapine, quetiapine, and ziprasidone, show benefits in decreasing oxidative stress through both reducing reactive oxygen species (ROS) formation and increasing oxidative protective factors, including glutathione and superoxide dismutase, leading to protection against apoptosis and myelin/oligodendrocyte loss [152].

### 4.3. Induction of Neurogenesis

Malignant gliomas grow in an infiltrative manner, thus damaging and disrupting the surrounding tissue. Drugs that can restore, even only partially, the damaged neural tissue can be beneficial for the patient. In this context, the role of antipsychotics in neuroprotection has been pointed out, especially for SGAs (see Table 1) that appear more effective than FGAs [152]. The neurogenic capability of antipsychotics has also been confirmed in vivo, where SGAs cause a 2- to 3-fold increase in the number of newly divided, NeuN-expressing, cells in the anterior SVZ of adult rats. Because NeuN is a neuronal marker, it is reasonable that antipsychotics may indeed act on neuronal progenitor cells [153].

## 5. Conclusions

This review illustrates how antipsychotics can be effective on several pathways central for neural stemness, development, function, metabolism and signal transduction, and how all the MoAs we describe for these drugs also appear functional in controlling cancer-cell growth, notably GBM. This tumor is known to display a noticeable heterogeneity in cell composition and possesses incredible plasticity, thus switching to a reprogrammed cell population according to the selective pressure generated by the current therapeutic approaches. This feature makes it predisposed to resistance to therapy [11,101]. Such a GBM biological behavior makes possible the option of using “dirty drugs”, i.e., compounds provided with pleiotropic and multifaceted MoAs (Figure 2), as antipsychotics definitely are. These drugs can hit widespread vulnerabilities of cancer cells and spare noncancer cells, due to a conceivably different toxicity pattern.

If we add that most of the medications described here are inexpensive when compared with the astounding costs of the new generation of cancer drugs, and that their times to reach the GBM patient’s bed can be drastically shortened, clinical trials involving the addition of selected antipsychotics to the state-of-art Stupp regimen should be particularly welcome. On the other hand, it should be considered that these drugs are not devoid of well-known and sometimes severe but drug-specific, dose-dependent and mostly reversible side effects, i.e., sedation and extrapyramidal syndrome [154].

Overall, repurposing of antipsychotics in GBM therapy could validly contribute to providing swift and inexpensive therapies for cancer patients since the impediments considered above are of lesser significance in the case of patients affected by cancers such as GBM, in which no second-line therapy is currently available or in those that have run through all known treatment opportunities.

## Figures and Tables

**Figure 1 cells-11-00263-f001:**
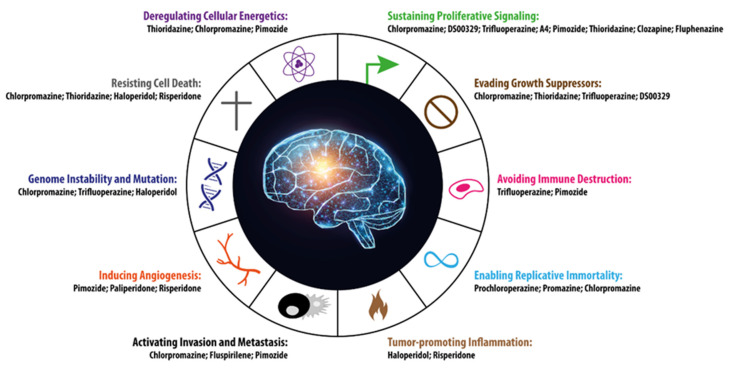
The 10 hallmarks of cancer identified by Hanahan and Weinberg [45] are represented along with the antipsychotic drugs potentially capable of interfering with each single specific cancer trait. The figure was inspired by [45] and modified appropriately.

**Figure 2 cells-11-00263-f002:**
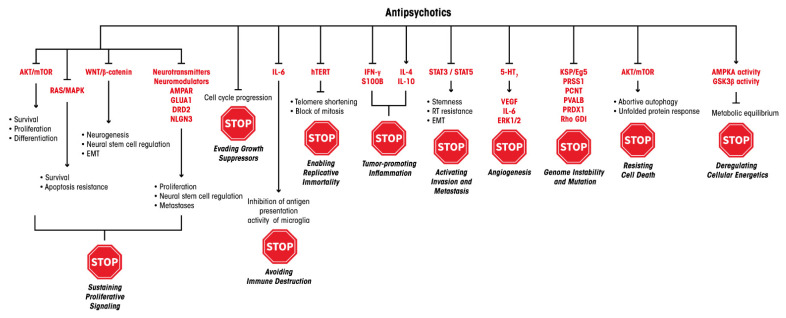
MoAs of antipsychotics and the molecular levels of their potential interference with the ten Hanahan and Weinberg’s hallmarks of cancer [45].

**Table 1 cells-11-00263-t001:** Summary of mechanism of action (MoA), indications and side effects of antipsychotic drugs.

Drug	MoA	Indications	Side Effects
**Typical antipsychotics**			
**(A) High potency:** Trifluoperazine, fluphenazine, haloperidol, thiothixene, fluopenthixol, pimozide.	- D2 receptor antagonists. (High potency > low potency)	- Schizophrenia (positive symptoms only, not first line); - psychotic disorders; - acute mania; - acute agitation; - Tourette syndrome; - Huntington disease.	Extrapyramidal Symptoms, neuroleptic malignant syndrome, hyperprolactinemia symptoms (galactorrhea in females and gynecomastia in males), QT prolongation, temperature instability (fluphenazine)
**(B) Low potency:** Chlorpromazine, thioridazine, sulpiride.	- H1, M1, α1 receptor antagonist (low potency > high potency).	- see high-potency indications.	Antimuscarinic side effects (e.g., dry mouth, constipation, blurred vision, urinary retention), orthostatic hypotension, sedation, chlorpromazine (corneal deposits), thioridazine (retinal deposits), cholestatic jaundice (chlorpromazine)
**(C) Mid-potency:** perphenazine, loxapine, prochlorperazine.	- intermediate D2 antagonists.	- antiemetic (prochlorperazine); - interact with lithium and potent CYP inducers (prochlorperazine); - see high-potency indications.	see high and low-potency side effects
**Atypical antipsychotics**			
Asenapine, ziprasidone, sertindole, zotepine, lurasidone, risperidone, paliperidone, iloperidone, sulpiride, olanzapine, quetiapine, clozapine.	- 5-HT_2A,_, H_1_, α_1,_, M_1_ antagonists; - D_2_ antagonist (lower affinity than typical antipsychotics).	Schizophrenia (first line), mania, Tourette syndrome, Obsessive–compulsive disorders, clozapine (for refractory schizophrenia only)	Metabolic syndrome (especially -pine), orthostatic hypotension, antimuscarinic side effects, sedation (especially quetiapine and clozapine), QT prolongation (especially ziprasidone), Extrapyramidal Symptoms and hyperprolactinemia (especially risperidone); specific for clozapine are agranulocytosis, myocarditis, seizures (dose-related) and wet pillow syndrome (rare)
aripiprazole, brexipiprazole.	- D2 partial agonist; - 5-HT_2A_ antagonist.	Schizophrenia (first line), mania, Tourette syndrome, obsessive–compulsive disorders	Lower risk of hyperprolactinemia and Extrapyramidal Symptoms, akathisia (aripiprazole), impulse-control disorder (aripiprazole)

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
