# Peer review of "Tackling the Behavior of Cancer Cells: Molecular Bases for Repurposing Antipsychotic Drugs in the Treatment of Glioblastoma"

_cells, 2022, doi:10.3390/cells11020263_

Round 1

Reviewer 1 Report

This review article is interesting. The authors discussed the antipsychotic drugs in glioblastoma treatment. There are some comments to the authors.

  1. Subtitle is suggested in the introduction.
  2. It is a good point to summarize the role of antipsychotic drugs at the ten cancer hallmarks. The authors analyzed the signal pathway in each hallmark. Simple mechanism figures to each pathway are recommended.
  3. Although the first line treatment of glioblastoma is radio therapy plus TMZ, other drugs approved by US FDA or Canada for glioblastoma, like Becacizumab, Carmustine or Lomustine could be mentioned (1,2).
  4. The discussion between target therapy and antipsychotic drugs is suggested.
  5. The clinical application should be discussed more.

References

  1. Weller, M., van den Bent, M., Preusser, M., Le Rhun, E., Tonn, J. C., Minniti, G., ... & Wick, W. (2021). EANO guidelines on the diagnosis and treatment of diffuse gliomas of adulthood. Nature Reviews Clinical Oncology18(3), 170-186.
  2. https://www.cancer.gov/about-cancer/treatment/drugs/brain

Author Response

We thank very much this reviewer for her/his positive comments and suggestions. We modified the manuscript accordingly:

  1. Many thanks for suggesting the addition of subtitles in the Introduction section. We did it, and now the text appears smoother.
  2. We thank this reviewer for the suggestion to add a figure in order to recapitulate the effects of antipsychotics on the ten hallmarks of cancer as identified by Hanahan & Weinberg. Now we added Figure 2 that describes the MoAs of antipsychotics and the molecular levels of their action.
  3. We thank the reviewer for this suggestion. According to her/his advice, we added the phrase “Currently, several drugs are utilized for recurrent GBM after first line treatment, e.g., bevacizumab, regorafenib, nitrosureas, but no standard treatments have been clearly identified for progressive disease”, adding the appropriate citations.
  4. Thanks for this suggestion. We ameliorated the discussion between target therapies, dirty drugs and antipsychotics in several points in the text.
  5. Thanks for this suggestion. According to her/his advice, we added/modified the paragraph “Moreover, the clinical benefit of Stupp protocol is correlated with the methylation status of the MGMT gene, a strong predictive biomarker of the response to temozolomide chemotherapy; indeed, only GBM patients with high methylation levels of MGMT have a survival advantage from this treatment. Thus, there is a great need of new research exploring the activity of new drugs, particularly in GBM patients carrying hypo- or un-methylated MGMT gene”, adding the appropriate citations.

Reviewer 2 Report

This review is a concise and systematic review on how specific antipsychotic drugs can be repurposed for targeting the different arms of the hallmarks of cancer as they apply to GBM and making the case for including these types of drugs in future clinical trials for GBM.

While the manuscript is overall well written some minor changes will add the readability:

An abbreviations list would facilitate reading.

For example, the definition of “THD” in line 137 does not appear in the text until line 335.

There seems to be an unfinished sentence in line 345 that states “unfolded protein response”, unless this is supposed to be the title of a subsection for the last paragraph of 3.9. In that case it should be labeled 3.9.1.

A sentence in this type of manuscript should not begin with “Anyway” (line 359). Consider, replacing with “However”

It might be useful to make an additional table with all the drugs mentioned throughout the manuscript, that includes their abbreviations (as mentioned in the manuscript), or perhaps include the abbreviations as part of table 1.

It seems that the word “neurogenetic” in line 434 is meant to be “neurogenic”.

The sentence in lines 445-449 needs re-writing for clarity.

Author Response

We thank very much this reviewer for her/his comments and suggestions. We modified the manuscript accordingly:

  • Thanks for suggesting the addition of an Abbreviations List, which has been added before the References section, according to Publisher suggestion. We also checked that each abbreviation has been mentioned at its first use in the text.
  • Thanks for noticing the error at line 345 of the previous version of the manuscript. The error has been corrected.
  • Thanks for suggesting to substitute “Anyway” at line 345 of the previous version of the manuscript with “However”. The text has been changed accordingly.
  • Thanks for suggesting to improve understanding of all abbreviations used in the text. As specified above, an Abbreviation List has been added.
  • Thanks for suggesting to substitute “neurogenetic” at line 434 of the previous version of the manuscript with “neurogenic”. The text has been corrected accordingly.
  • Thanks for the suggestion to rephrase the sentence in lines 445-449. We hope that now it is more understandable.

Round 2

Reviewer 1 Report

No further questions for the authors.